# Fully Dynamic Coreset Spectral Clustering

Ben Jourdan [1]   Peter Macgregor [2]   Gregory Schwartzman [3]

## Abstract

We present a fully dynamic data structure that supports edge and node updates and cluster membership queries for the Normalised Cut problem with strong theoretical guarantees. Furthermore, our data structure outperforms the state of the art significantly on real world datasets. At the heart of our data structure is the novel notion of *Just-in-Time Sampling Trees*. The worst-case edge update time of our data structure is $O(\log n)$ where $n$ is the number of nodes in the current graph. Let $d_{\max}$ be the maximum degree of the current graph, let $T_{NC}(n', k')$ be the running time of an $\alpha$-approximation algorithm for the Normalised Cut problem on $n'$ vertices and $k'$ clusters, and let $\mathrm{vol}(Y)$ be the sum of the unweighted degrees of all nodes in a set $Y$. The worst-case query time of our data structure to label all nodes in $Y$ is $O\left(kd_{max}^2 \log(n)\right) + \gamma(n, \varepsilon, k, Y)$, with approximation ratio $\alpha \frac{1+\varepsilon}{1-\varepsilon}$, where $\gamma(n, \varepsilon, k, Y)$ is defined to be $O(\log(n) \log^\star(n) \varepsilon^{-4} k^2 + T_{NC}(\varepsilon^{-4}k^2, k) + \varepsilon^{-8}k^4 + \min\{|Y| k^2 \varepsilon^{-4}, \mathrm{vol}(Y)\})$. Assuming $d_{\max}$ is polylogarithmic, as is the case with many sparse real-world graphs, our method achieves the best known trade-off between query time and update time.

## 1. Introduction

Clustering large-scale graphs is central to understanding the structure of modern data, but real-world graphs are rarely static. Social interactions, communication patterns, and information flows evolve continuously, rendering static clustering methods inadequate. To capture these changing structures, we need algorithms that can efficiently adapt to dynamic graphs while preserving the theoretical and practical strengths of spectral clustering.

Among the many approaches to graph clustering, spectral clustering has stood out for its ability to uncover complex, non-linear structures in data. Closely related methods such as kernel $k$-means share this strength, and together they have found wide application across domains ranging from medical research to network science (Gönen & Margolin, 2014; Kuo et al., 2014; White & Smyth, 2005). These successes make spectral methods a natural starting point when extending clustering algorithms to the dynamic setting.

Given a *Euclidean* dataset $X$, both kernel $k$-means and spectral clustering rely on a kernel similarity function $K : X \times X \to \mathbb{R}_{\geq 0}$, often represented as an $n \times n$ matrix, where $n = |X|$. Spectral clustering interprets this matrix as the adjacency matrix of a *similarity graph* and seeks to minimise the *normalised cut* objective (defined in Appendix A) (Von Luxburg, 2007). In contrast, kernel $k$-means leverages the fact that the kernel implicitly defines an embedding $\phi : X \to \mathcal{H}$ into a Hilbert space, with

$$\langle \phi(x), \phi(y) \rangle = K(x, y), \quad \forall x, y \in X. \tag{1}$$

The clustering problem is then to minimise the $k$-means objective in this space, typically solved using a kernelized version of Lloyd's algorithm (Dhillon et al., 2004).

A key observation, due to Dhillon et al. (2004), is that the normalised cut and kernel $k$-means objectives are equivalent up to constant factors when written as trace minimisation optimisation problems. In particular, optimising the normalised cut on a graph $G(V, E)$, the average conductance of a $k$-partition, can be reformulated as an instance of kernel $k$-means with a suitable kernel, which we refer to as the *graph kernel* (Definition 3). The other direction also holds; any kernel $k$-means instance can be reformulated as a normalised cut instance. This duality establishes a deep connection between the two approaches.

Despite this equivalence, the two methods have historically been studied in isolation, with algorithmic advances developed separately for each. A recent result by Jourdan et al. (2025) showed how to bridge this gap: they introduced the first coreset[1] spectral clustering algorithm. This algorithm

---

[1]Department of Informatics, University of Edinburgh, United Kingdom [2]School of Computer Science, University of St Andrews, United Kingdom [3]Graduate School of Advanced Science and Technology, JAIST, Ishikawa, Japan. Correspondence to: Ben Jourdan <ben@jourdan.me>.

*Proceedings of the 43 $^{rd}$ International Conference on Machine Learning*, Seoul, South Korea. PMLR 306, 2026. Copyright 2026 by the author(s).

---

[1]A coreset is a small, weighted subset of the input that approximates the original dataset with respect to a given objective up to a

clusters a *coreset graph* (an edge reweighted induced subgraph) to infer a good labelling of the original graph by exploiting the equivalence with kernel $k$-means. Their analysis moves between graph space and kernel space, yielding a framework where a coreset for kernel $k$-means can be used to accelerate spectral clustering—achieving significant speed-up when the coreset is small. We summarize their result informally below (with full details deferred to Appendix A).

**Theorem 1** (Jourdan et al. (2025) (informal version of Theorem 3)). *Given a weighted graph $G$ and a coreset $S$ for the graph kernel of $G$, normalised cut on $G$ with $k$ clusters can be approximately solved in time $O(T_{NC}(|S|, k) + k^4 + \text{vol}(G))$.*

**Dynamic Clustering** Before we state our results, let us intuitively define the dynamic clustering problem. Given a dynamic weighted graph $\mathcal{G} = G_1, G_2, ...$ (this sequence can be either finite or infinite) where $G_i, G_{i+1}$ only differ by at most a single edge, our goal is to maintain a data structure that supports the following operations: edge insertions and deletions[2], and querying the cluster memberships of a set of points; given a set $Y \subseteq X$ and parameter $k$, we return for every $x \in Y$ a cluster in $\{1, \ldots, k\}$. The goal is to design a data structure where both the edge updates and cluster queries are as fast as possible while maintaining a good quality of clustering (see Section 5 for details of how this is measured). Note that graph parameters such as the number of nodes, number of edges, average degree and maximum degree are not fixed throughout the evolution of the graph. When stating these values, they always refer to a specific $G_i$, and this $i$ should be clear from context. For example, if an edge update time is $O(\log n)$ then $n$ refers to the number of nodes at the time of the update.

We note that Theorem 1 implies a framework for dynamic spectral clustering. Given a dynamic graph, if we can dynamically maintain a coreset, $S$, for the graph kernel, we would have a natural dynamic algorithm for spectral clustering with query time $O(T_{SC}(|S|, k) + |S|^2 + \text{vol}(Y))$ for a node set $Y$. While we present the results of Jourdan et al. (2025) in Appendix A for completeness, in the main body of the paper we focus on dynamically maintaining a coreset for the graph kernel.

## 1.1. Our Results

We present a fully dynamic data structure for spectral clustering, which is significantly faster compared to existing approaches. Our data structure builds upon the static al-

gorithm of Jourdan et al. (2025). We briefly outline their results. The heart of the static coreset algorithm repeatedly samples a subset of nodes according to two adaptive distributions. By adaptive, we mean that the sampling distribution changes during sampling. To construct the coreset, they first sample an auxiliary set of nodes according to one adaptive distribution. Then, they sample the final coreset according to the other distribution, derived from the sampled auxiliary set. To allow for efficient sampling, a *sampling tree* is used. This is a data structure that is constructed once before the coreset algorithm is executed, and is updated throughout the execution of the algorithm (to support the adaptive nature of the sampling). The initial construction time of the sampling tree is *linear* in the size of the input; rebuilding a sampling tree from scratch in the dynamic setting is prohibitive.

**Our Approach** Adapting this algorithm directly to the dynamic setting would require reconstructing the sampling tree for every change in the input graph, taking linear time. Instead, we design a dynamic data structure to maintain the sampling tree under edge insertions and deletions. This allows us to achieve a logarithmic update time per edge update. To query clusters, we 1) use the tree to quickly sample a coreset, recall that the coreset is tiny compared to the full graph, 2) cluster the coreset, and 3) use the coreset clusters to determine the cluster memberships of the query nodes. Jourdan et al. (2025) maintain the probabilities in the sampling trees explicitly, which does not allow for efficient updates when the graph changes. A naive attempt would require updating all the nodes in the sampling tree after every update, which takes linear time.

Instead of maintaining the probabilities directly in the sampling tree, our data structure has each node maintain auxiliary quantities which allows it to quickly compute the sampling probabilities on demand. Our approach is significantly faster both in theory and in practice on sparse graphs. We compare our data structure to other dynamic approaches and naive baselines in Table 1. When queried, Naive simply runs the full spectral clustering algorithm on the entire graph and returns the node cluster. Static uses coreset spectral clustering instead (Jourdan et al., 2025). We also compare to the dynamic spectral clustering data structure of Laenen & Sun (2024), which only supports edge additions, but not deletions, and the Merge&Reduce data structure that combines the merge and reduce framework of Henzinger & Kale (2020) with the static coreset algorithm of Jourdan et al. (2025). The Naive and Static approaches achieve very fast edge update times, but their node query time is prohibitively slow. Merge&Reduce is the only data structure to achieve a competitive query time compared to our approach, however its edge update time is prohibitively slow in practice. We observe experimentally that our edge update time is much faster than Merge&Reduce while the query time is similar for large graphs.

---

multiplicative factor close to 1.

[2]Node insertions and deletions are also supported implicitly; inserting $(u, v)$ will add a node if either $u$ or $v$ is not already in the graph, and removing all edges adjacent to a node $v$ removes it from the graph.

| Algorithms | Edge Update Time | Query Time for set $Y \subseteq V$ | Approx. |
|---|---|---|---|
| **Ours** | $O(\log n)$ | $O(kd_{max}^2 \log n)$ $+ \gamma(n, \varepsilon, k, Y)$ | $\alpha\frac{1+\varepsilon}{1-\varepsilon}$ |
| Laenen & Sun | $O(1)$ amortized | $O(n/\log n)$ amortized | N/A |
| Merge&Reduce | $O(\varepsilon^{-4}k^3 \log^5 n)$ | $T_{NC}(\varepsilon^{-4}k^2, k)$ $+ O(\varepsilon^{-8}k^4$ $+ \min\{|Y| \, k^2\varepsilon^{-4}, \mathrm{vol}(Y)\})$ | $\alpha\frac{1+\varepsilon}{1-\varepsilon}$ |
| Naive | $O(1)$ amortized | $T_{NC}(n, k)$ | $\alpha$ |
| Static | $O(1)$ amortized | $O(nk \log n)$ $+ \gamma(n, \varepsilon, k, Y)$ | $\alpha\frac{1+\varepsilon}{1-\varepsilon}$ |

*Table 1.* Comparison of data structures in terms of edge update and node set query time. We define $\gamma(n, \varepsilon, k, Y)$ to be $O(\log(n) \log^\star(n)\varepsilon^{-4}k^2 + T_{NC}(\varepsilon^{-4}k^2, k) + \varepsilon^{-8}k^4 + \min\{|Y| \, k^2\varepsilon^{-4}, \mathrm{vol}(Y)\})$. Unless stated otherwise, running times are worst-case. The method of Laenen & Sun (2024) targets a different objective ($k$-way expansion) to the others so it is not possible to compare approximation ratios. They also do not provide the effect of $k$ on update and query times. The values $n, d_{max}, d_{avg}, \mathrm{vol}(Y)$ are the number of nodes, maximum degree, average degree and volume of $Y$ w.r.t the dynamic graph at the time of edge update / node set query. $T_{NC}(n, k)$ is the running time of an $\alpha$-approximation algorithm for the normalised cut problem 16 on an $n$-vertex graph with $k$ clusters. See Appendix E for a derivation of these running times.

Our result is significant because we are able to efficiently maintain a dynamic coreset whose size doesn't depend on even logarithmic factors of $n$, while achieving superior worst-case running times compared to existing dynamic data structures. By the nature of the Merge&Reduce method, the update time for Merge&Reduce must incur a polynomial dependence on $k$. Our method avoids this dependence and as a consequence, $k$ can be chosen at query time. This makes it much more flexible in practice where it is typical for the number of clusters to change over time.

**Theorem 2** (Main result (informal version of Theorem 6)).
*There exists a dynamic data structure that maintains an $\alpha\frac{1+\varepsilon}{1-\varepsilon}$-approximate normalised cut that supports edge insertions and deletions and has an edge update time of $O(\log n)$ and a set query time of $O(kd_{max}^2 \log(n) + \gamma(n, \varepsilon, k, Y))$ when queried with a node set $Y$ and number of clusters $k$.*

In the analysis of Theorem 6, we require an algorithm that achieves an $\alpha-$approximation algorithm for the Normalised Cut problem. We focus on spectral clustering because of its relationship to the normalised cut problem; in practice, other clustering algorithms may be preferred.

**Paper Structure**    We review related work in Section 1.2. In Section 2 we define notation and provide formal definitions. In Section 3 we overview the static coreset algorithm

of Jourdan et al. (2025), setting the groundwork for our dynamic data structure. In Section 4 we present our dynamic coreset spectral clustering data structure. In Section 5 we experimentally evaluate our data structure against all data structure in Table 1. A formal version of Theorem 2 appears in Appendix F.

## 1.2. Related Work

Prior dynamic spectral clustering methods (Dhanjal et al., 2014; Martin et al., 2018; Ning et al., 2007) largely track changes of approximate eigenvectors under perturbations, typically without explicit approximation guarantees for the returned clusters and often with a fixed vertex set. The state of the art for the *incremental* setting is due to Laenen & Sun (2024): it considers *insertions only* (edges and possibly new vertices), and assumes a *dynamic gap / cluster-structure* condition—roughly, that at designated times the graph admits well-separated $k'$ clusters while each update is an edge insertion introducing at most one new vertex. Under these assumptions, they achieve $O(1)$ amortised update and near linear[3] amortised query time with provable approximation guarantees for the output clusters.

In (Henzinger & Kale, 2020) a generic approach for turning static coresets into dynamic coresets is presented. This is applied to k-means, followed by both practical and theoretical improvements (Henzinger et al., 2024; la Tour et al., 2024). Recently, Jourdan et al. (2025) showed how to use coresets together with the equivalence between Kernel $k$-means and spectral clustering (Dhillon et al., 2004) to design a Coreset Spectral Clustering algorithm—achieving the same approximation guarantees as spectral clustering (up to a small multiplicative error). This combined with the results of Henzinger & Kale (2020) gives rise to a dynamic spectral clustering data structure that supports both edge additions and deletions. This results in a polylogarithmic update time and a *sublinear* query time, while maintaining the approximation guarantees of Jourdan et al. (2025).

## 2. Preliminaries

Let $X$ be a set of $n$ objects, and $K : X \times X \to \mathbb{R}_{\geq 0}$ be a function measuring the pairwise similarity of data points in $X$. Let $\phi : X \to \mathcal{H}$ be the function implicitly defined by $K$ that maps data points in $X$ to the unique Hilbert space such that $\langle \phi(x), \phi(y) \rangle = K(x, y)$ for all $x, y \in X$. The function $K$ is usually represented as a positive semi-definite matrix: if $\Phi = [\phi(x_1), \ldots, \phi(x_n)]$, then $K = \Phi^T \Phi \in \mathbb{R}^{n \times n}$ with $K_{ij} = \langle \phi(x_i), \phi(x_j) \rangle$. Let $\Delta(x, y) \triangleq \|\phi(x) - \phi(y)\|^2$ denote the squared distance in feature space for all $x, y \in X$, and $\Delta(x, C) = \min_{c \in C} \Delta(x, c)$ denote the smallest

---

[3]Strictly speaking, the amortised query time is $O(n/\log n)$ where $n$ is the number of nodes at query time.

squared distance from $x \in X$ to a set $C \subseteq X$. We refer to the diagonal elements of $K$ as self similarities and say $x$ is a neighbour of (or incident to) $y$, written $x \sim y$, iff $\langle \phi(x), \phi(y) \rangle \neq 0$. If $x \in X$ and $S \subset X$, we say $x$ is incident to $S$ if and only if $x$ is incident to at least one element of $S$.

We make use of the following concepts in our analysis.

**Definition 1** (kernel $k$-means objective). *Given a weighted dataset $X$ with weights $w : X \rightarrow \mathbb{R}_+$ and feature map $\phi : X \rightarrow \mathcal{H}$ satisfying equation 1 for some kernel function $K$, the weighted kernel $k$-means objective with respect to an arbitrary set of points $C \subseteq \mathcal{H}$ is $\text{COST}_w(X, C) = \sum_{x \in X} w(x) \Delta(x, C)$.*

**Definition 2** ($\varepsilon$-coresets). *For $0 < \varepsilon < 1$, an $\varepsilon$-coreset for kernel $k$-means on a weighted dataset $X$ with weights $w : X \rightarrow \mathbb{R}_+$ is a reweighted subset $S \subseteq X$ such that for the Hilbert space $\mathcal{H}$ and map $\phi : X \rightarrow \mathcal{H}$ satisfying Equation 1, we have $\text{COST}_{w'}(S, C) \in (1 \pm \varepsilon) \cdot \text{COST}_w(X, C), \quad \forall C \subset \mathcal{H}$ with $|C| = k$. where $w' : S \rightarrow \mathbb{R}_+$ gives the weight for each element in the coreset.*

**Definition 3** (Graph kernel, (Dhillon et al., 2007)). *Given a graph with positive edge weights, graph $G = (V, E)$, $k$ with adjacency matrix $A$ and degree matrix $D$ let $K = D^{-1}AD^{-1} + \sigma D^{-1}$ and $W = D$, where $\sigma$ is chosen such that $K$ is positive definite. We call $(K, W)$ the graph kernel for $G$. We say that a reweighted subset $V' \subseteq V$ is an $\varepsilon$-coreset for the graph kernel of $G$ if its an $\varepsilon$-coreset for kernel matrix $K$ and weight matrix $W$. The diagonal entries of $W$ correspond to the values of $w$ in Definition 1. We refer to the edge weight between two nodes in $G$ as $w(x, y)$. If $x \sim y$ then $w(x, y) > 0$, otherwise $w(x, y) = 0$.*

**Definition 4** (Seed sets and seed set weight). *Let $X$ be a set of points with weights $w : X \rightarrow \mathbb{R}_{\geq 0}$ and $C \subseteq X$ be a set of seeds. Then for any $x \in X$, we define the seed set of $x$ with respect to $C$ to be the points in $X$ that share the same closest seed as $x$. That is, we define the seed set of $x$ with respect to $C$ to be $C(x) \triangleq \{y \in X | \arg\min_{c \in C} \Delta(x, c) = \arg\min_{c \in C} \Delta(y, c)\}$ where ties are broken arbitrarily. We define the weight of a seed set to be the sum of the weight of the points in the seed set. That is, $w(C(x)) \triangleq \sum_{y \in C(x)} w(y)$.*

## 3. Static Coreset Spectral Clustering

Given an input graph, the algorithm of Jourdan et al. (2025) first extracts the corresponding weighted kernel $k$-means problem via the equivalence to the normalised cut problem, and then constructs an $\varepsilon$-coreset. Following this, again via the equivalence, they solve the corresponding normalised cut problem on the coreset graph to get the coreset partition. Finally, they label the rest of the data by considering kernel distances to the implied centers induced by the coreset graph partition. A full description of their algorithm, accompanied

by an intuitive illustration (Figure 3), appears in Appendix A. Going forward we focus on the coreset construction at the heart of their algorithm.

The algorithms of Jourdan et al. (2025) use the following notation. For dataset $X$ and seed set $C \subseteq X$, for $x \in X$, and $S \subseteq X$, we define the following:

$$f(x, C) \triangleq w(x)\Delta(x, C), \qquad f(S, C) \triangleq \sum_{z \in S} f(z, C),$$

$$g(x, C) \triangleq \frac{f(x, C)}{f(X, C)} + \frac{w(x)}{w(C(x))}, \quad g(S, C) \triangleq \sum_{z \in S} g(z, C).$$

Intuitively, $f(x, C)$ is the *weighted* distance between $x$ and $C$ in feature space, $g(x, C)$ is the sum of the relative weight of $x$ w.r.t. its seed set $\left( \frac{w(x)}{w(C(x))} \right)$ and the relative weighted distance of $x$ to $C$ w.r.t. all points $\left( \frac{f(x, C)}{f(X, C)} \right)$.

**Constructing Coresets** The $\varepsilon$-coreset Algorithm of Jourdan et al. (2025) is based on the algorithm of Jiang et al. (2024), augmented for sparse graphs (Algorithm 1). It consists of multiple rounds of importance sampling that progressively reduce the size of the input. Each round seeds the importance of each point using the distribution given by running $D^2$-sampling (Algorithm 3), and then *smoothing* this distribution according to the weight of the point, normalised by the weight of its seed set (line 4 of Algorithm 2).

Each round approximately reduces the number of unique elements to be logarithmic in the number of unique elements in the previous round until the number of unique elements does not decrease. In practice, one round is usually enough.

**Sampling Trees** To implement the sampling in Algorithm 3 efficiently for a graph kernel, Jourdan et al. (2025) make use of *sampling trees* (Wong & Easton, 1980). Sampling trees provide an efficient way to sample elements from a set $X$ according to some unnormalised distribution as well as updating the distribution. A sampling tree is a balanced tree where the leaves correspond to elements of $X$ and internal nodes correspond to the union of their children. For sampling, leaves store unnormalised probabilities and internal nodes store the sums of their children's unnormalised probabilities. For more details, refer to Appendix B.

---

**Algorithm 1** $\varepsilon$-coreset for kernel $k$-means on dataset $X$ with kernel $K$ (Jiang et al., 2024)

---

1: **Input:** $X_0 \leftarrow X, i \leftarrow 0$
2: **repeat**
3:      $i \leftarrow i + 1$ and $\varepsilon_i \leftarrow \varepsilon / (\log^{(i)} \|X_0\|_0)^{1/4}$   $\triangleright \log^{(i)}(\cdot)$ is the $i$th iterated logarithm.
4:      $X_i \leftarrow \text{IMPORTANCE-SAMPLING}(X_{i-1}, \varepsilon_i)$ $\triangleright$ Alg. 2
5: **until** $\|X_i\|_0$ does not decrease compared to $\|X_{i-1}\|_0$
6: **return** $X_i$

---

**Algorithm 2** Importance-Sampling$(X, \varepsilon)$

1: Let $C^* \leftarrow D^2\text{-Sampling}(X)$  $\qquad\qquad$ ▷ Alg. 3
2: $\forall x \in X, \sigma_x \leftarrow g(x, C^*)$
3: $\forall x \in X, p_x \leftarrow \frac{\sigma_x}{\sum_{y \in X} \sigma_y}$
4: Draw $N \leftarrow O\left(\frac{k^2 \log^2(k) \log(\|X\|_0)}{\varepsilon^4}\right)$ i.i.d. samples from $X$, using $(p_x)_{x \in X}$
5: Let $D$ be the sampled set; for each $x \in D$ let $w_D(x) \leftarrow \frac{w(x)}{p_x N}$
6: **return** weighted set $D$

---

**Algorithm 3** Modified $D^2\text{-Sampling}(X)$

1: $x^* \leftarrow \arg\min_{x \in X} \langle \phi(x), \phi(x) \rangle$
2: $C \leftarrow \{x^*\}$
3: Draw $x \in X$ uniformly at random
4: $C \leftarrow C \cup \{x\}$
5: **for** $i = 1, \ldots, k-1$ **do**
6: $\quad$ Draw $x \in X$, using probabilities $\frac{f(x,C)}{f(X,C)}$
7: $\quad$ $C \leftarrow C \cup \{x\}$
8: **end for**
9: **return** $C$

---

As written, it appears that sampling proportional to $\sigma_x$ on line 2 of Algorithm 2 requires us to recompute a full sampling tree every time we build a coreset, since $\sigma_x$ depends on $C^*$ which is the output of Algorithm 3. To overcome this, we build a unified data structure that fuses Algorithm 2 and Algorithm 3 so that we can quickly sample proportional to $\sigma_x$, under updates to the graph. This will correspond to maintaining a single sampling tree for $f(x, \{x^*\})$ and $g(x, \{x^*\})$ where $x^*$ is the node in the graph with highest degree. Sampling the rest of $C$ in Algorithm 3 can then be performed quickly. As we sample the rest of $C$, we efficiently update the sampling tree to respect $f(x, C)$ and $g(x, C)$ after every new point is added to $C$. We use timestamps to lazily reset the state of the sampling tree when we begin each query.

## 4. Just-in-Time Sampling Trees for Dynamic Graph Kernel Coresets

Our goal is to maintain a sampling tree and associated data structures so that we can quickly run a fused version of Algorithm 2 and Algorithm 3. In this section, we present such an algorithm that runs in time $O(d_{max}^2 \log(n))$ time. Under edge updates, updating the associated data structures takes $O(\log(n))$ time. Combined with the running time analysis in Appendix E and Theorem 3, this immediately implies Theorem 6. We must be able to efficiently sample from and update the sampling tree according to the following unnormalised distributions: 1) $f(x, C) = w(x) \cdot \Delta(x, C)$, line 3 of Algorithm 3, 2) $g(x, C^*) = \frac{w(x) \cdot \Delta(x, C^*)}{\text{COST}_w(X, C^*)} + \frac{w(x)}{w(C^*(x))}$, line 2 of Algorithm 2.
Naively, we could try to build and maintain a sampling tree where leaf nodes store $f(x, \{x^*\})$ or $g(x, \{x^*\})$ and

internal nodes store the sum of their children. Then, when we need to compute a coreset, we simulate $D^2$-sampling by updating the tree to compute intermediate values of $f(S, C)$ and $g(S, C)$. However, as seeds are added to the set $C$ in Algorithm 3 or edges are added or removed from the graph, a linear number of values in the sampling tree may need to be changed. For example, after a series of edge updates, the node with highest degree $x^*$ may change, which would require a contribution update for every leaf in the tree. Instead of storing and updating $f(x, \{x^*\})$ or $g(x, \{x^*\})$ directly, each node maintains values that can be used indirectly to compute them in a *just-in-time manner*. Only when actually sampling from the tree will the values of internal nodes be computed. Of the values that we do store, only $O(\log n)$ updates to the sampling tree are required under graph updates and only $(d_{max}^2 \log n)$ updates are required to simulate a single iteration of $D^2$-sampling.

**Maintaining** $f(S, \{x^*\})$ **And** $g(S, \{x^*\})$  $\quad$ Let $G = (V, E, w)$ be an undirected graph on $n$ vertices after a sequence of edge insertions/deletions and let the (dynamic) kernel matrix be $K = D^{-1}AD^{-1} + \sigma D^{-1}$ and $W = D$ where $\sigma$ is a parameter set to make sure $K$ is positive definite (Definition 3). Accordingly, for any $x, y \in X$,

$$\langle \phi(x), \phi(x) \rangle = K(x, x) = \frac{\sigma}{\deg(x)},$$

$$\langle \phi(x), \phi(y) \rangle = K(x, y) = \frac{w(x, y)}{\deg(x) \deg(y)}.$$

To quickly run a fused version of Algorithms 2 and 3 in the presence of edge updates, we maintain a sampling tree whose nodes, each representing a set $S \subseteq X$, can quickly compute $f(S, \{x^*\})$ and $g(S, \{x^*\})$. This can be thought of as maintaining $f(S, \{x^*\})$ and $g(S, \{x^*\})$ up to line 3 in Algorithm 3. To maintain $x^*$ itself, we use a max-heap over the degrees of the vertices in $G$ since $x^*$ corresponds to a node with maximal degree, regardless of the value of $\sigma$. In Sections 4.1 and 4.2, we will see what each node in the sampling tree actually needs to maintain to compute $f(S, C)$ and $g(S, C)$ just-in-time as $C$ grows with each iteration of Algorithm 3, starting with $C = \{x^*\}$.

### 4.1. Computing $f(S, C)$ **Just-In-Time.**

Now assume that $x^*$ has been added to the seed set $C$ in line 3 of Algorithm 3. Then, for any $x \in X$, we can derive the following expression for $f(x, C)$ in terms of $C$.

$$f(x, C)$$
$$= \begin{cases} w(x)\langle \phi(x), \phi(x) \rangle + w(x)\langle \phi(x^*), \phi(x^*) \rangle & x \not\sim C, \\ w(x)\langle \phi(x), \phi(x) \rangle + w(x) \min_{c \in C} \Big( \langle \phi(c), \phi(c) \rangle & x \sim C, \\ \qquad\qquad\qquad\qquad -2\langle \phi(x), \phi(c) \rangle \Big) \end{cases}$$

$$= \begin{cases} \sigma + \deg(x)\Big(\dfrac{\sigma}{\deg(x^*)}\Big) & x \not\sim C, \\[2mm] \sigma + \deg(x) \min_{c \in C} \Big(\dfrac{\sigma}{\deg(c)} & x \sim C, \\[2mm] \qquad -2\dfrac{w(x,c)}{\deg(x)\deg(c)}\Big) \end{cases} \quad (2)$$

with the relation that for all $x \in X$ and $C \subset X$ where $x^* \in C$ it holds that

$$f(x, C \cup \{y\}) = \begin{cases} f(x,C) & x \not\sim y \\[2mm] \min\Big(f(x,C), w(x)\Delta(x,y)\Big) & x \sim y \end{cases}$$
$$\qquad (3)$$

**Easy Case.** Suppose $S \subset X$ is a set such that for all $x \in S$, we have $x \not\sim C$. That is, none of the nodes represented by $S$ have any edges to nodes in $C$. This corresponds to every point being in the first case of (2). Accordingly, we get the following expression for $f(S,C)$:

$$f(S,C) = \sum_{x \in S} f(x,C) = \sum_{x \in S}\Big[\sigma + \deg(x)\Big(\frac{\sigma}{\deg(x^*)}\Big)\Big]$$
$$= \sigma\,|S| + \frac{\sigma}{\deg(x^*)}\sum_{x \in S}\deg(x) \qquad (4)$$

From (4), it's clear that each node in the sampling tree should maintain $|S|$ and $\sum_{x \in S}\deg(x)$. If each node maintains this information, then given an arbitrary value of $\sigma$ and choice of $x^*$, it can successfully compute $f(S,C)$ according to (4) in constant time. Crucially, both terms decompose linearly; suppose $S \subseteq X$ is represented by a node $L$ in the sampling tree and $U, V$ form a partition of $S$ and are represented by the child nodes of $L$. Then the values of $|S|$ and $\sum_{x \in S}\deg(x)$ can be computed by summing these values for $U$ and $V$. As such, maintaining this information in the sampling tree under edge updates can be accomplished in $O(\log n)$ time.

**General Case.** For $S \subset X$ which contain nodes with edges to $C$, we track the difference between the expression in (4) and the true value of $f(S,C)$. To do this, we decompose $f(S,C)$ into a base part, $f^b(S)$ corresponding to the situation where there are no edges to $C$, and a delta term $f_\delta(S,C)$, which stores the difference to the true value of $f(S,C)$. For all $x \in X$ and $S \subseteq X$, we define,

$$f^b(x) \triangleq \sigma + \deg(x)\Big(\frac{\sigma}{\deg(x^*)}\Big), \quad f^b(S) \triangleq \sum_{x \in S}f^b(x),$$
$$f_\delta(x,C) \triangleq f^b(x) - f(x,C), \quad f_\delta(S,C) \triangleq \sum_{x \in S}f_\delta(x,C).$$

From the definitions, we have that for any $S \subseteq X$

$$f^b(S) - f_\delta(S,C) = \sum_{x \in S}f^b(x) - \Big(f^b(x) - f(x,C)\Big)$$
$$= \sum_{x \in S}f(x,C) = f(S,C). \qquad (5)$$

Accordingly, we need to store and update $f_\delta(S,C)$ for every internal node of the sampling tree.

**Maintaining $f_\delta(S,C)$.** When we begin Algorithm 3, all the delta terms will initially be zero. Every time we add a node $y$ to the seed set $C$, from (3), it suffices to check whether $f(x,\{y\}) < f(x,C)$ for any neighbour in $\{x \in X | x \sim y\}$. For those neighbours where $f(x,y) < f(x,C)$, we update their respective delta terms in the leaves of the sampling tree:

$$f_\delta(x, C \cup \{y\}) = f^b(x) - f(x,y), \qquad (6)$$

and we add the difference in delta term to every internal node's delta term along the path from the leaf node representing $x$ to the root of the sampling tree:

$$f_\delta(S, C \cup \{y\})$$
$$= f_\delta(S,C) + \Big(f_\delta(x, C \cup \{y\}) - f_\delta(x,C)\Big). \quad (7)$$

Since we sample $k$ seeds in Algorithm 3, each with at most $d_{max}$ neighbours, the running time to maintain $f(x,C)$ while simulating a single round of Algorithm 3 is $O(k \cdot d_{max}\log n)$.

**Logical Timestamps For Stale Delta Terms.** As stated, we will need to reset all the non-zero delta terms to zero before running Algorithm 3 again for the next query. While this would only incur a constant factor in the running time, we can use a simple trick to avoid this. We maintain a global logical timestamp $T$ that we increment before starting Algorithm 3 and also store timestamps at every node in the sampling tree. When we try to read a delta term, if that node's timestamp doesn't match $T$, the delta term is stale so we reset it to zero and update the timestamp for that node. Otherwise we read the value as is. This ensures that we forget about delta terms from previous calls to Algorithm 3.

### 4.2. Computing $g(S,C)$ **Just-In-Time.**

The difficulty with computing the $g(x,C)$ terms is $w(C(x))$, Definition 4, in the denominator of $\frac{w(x)}{w(C(x))}$ in the definition of $g(x,C)$. Recall that $C(x) = \{y \in X | \arg\min_{c \in C}\Delta(x,c) = \arg\min_{c \in C}\Delta(y,c)\}$ is the seed cluster that $x$ belongs to, and $w(C(x)) = \sum_{y \in C(x)}w(y)$ is the weight of the seed cluster that $x$ belongs to. Initially, after line 3 of Algorithm 3, for all $x \in X$, we have $C(x) = C(x^*) = X$ and $w(C(x)) = w(C(x^*)) = \sum_{y \in X}w(y)$. This is because there is only one seed, $x^*$ in $C$. As more seeds are added to $C$, points in $C(x^*)$ will move to different seed clusters and so $w(C(x^*))$ will decrease. Likewise, vertices in other seed sets may also move to newer seed sets. As a result, the addition of a new seed may affect the denominator of $\frac{w(x)}{w(C(x))}$ for potentially $\Omega(n)$ terms. We need a way of managing this efficiently, similarly to the case for the $f(x,C)$ terms. We do this by having every node in the sampling tree track a decomposition of $\sum \frac{w(x)}{w(C(x))}$ over the

vertices that they represent. To aid our decomposition, we use the following notation. For any $x \in X$ and $S \subseteq X$, define

$$h(x, C) \triangleq \frac{w(x)}{w(C(x))}, \tag{8}$$

$$h(S, C) \triangleq \sum_{y \in S} \frac{w(y)}{w(C(y))}, \tag{9}$$

$$h^b(x, C) \triangleq \begin{cases} w(x) & x \in C(x^*) \\ 0 & x \notin C(x^*), \end{cases} \tag{10}$$

$$h^b(S, C) \triangleq \sum_{x \in S} h^b(x, C) = \sum_{x \in S \cap C(x^*)} w(x), \tag{11}$$

$$h_s(x, C) \triangleq \begin{cases} 0 & x \in C(x^*) \\ \frac{w(x)}{w(C(x))} & x \notin C(x^*), \end{cases} \tag{12}$$

$$h_s(S, C) \triangleq \sum_{x \in S} h_s(x, C) = \sum_{x \in S \cap (X \setminus C(x^*))} \frac{w(x)}{w(C(x))} \tag{13}$$

The decomposition is then $h(x, C) = \frac{h^b(x, C)}{w(C(x^*))} + h_s(x, C)$, so that

$$g(x, C) = \frac{f(x, C)}{f(X, C)} + h(x, C)$$

$$= \frac{f(x, C)}{f(X, C)} + \frac{h^b(x, C)}{w(C(x^*))} + h_s(x, C),$$

$$g(S, C) = \frac{f(S, C)}{f(X, C)} + h(S, C) \tag{14}$$

$$= \frac{f(S, C)}{f(X, C)} + \frac{h^b(S, C)}{w(C(x^*))} + h_s(S, C) \tag{15}$$

Given this decomposition of $g(S, C)$, it suffices to maintain $w(C(x^*))$ outside of the sampling tree and have every node maintain $h^b(S, C)$ and $h_s(S, C)$. We additionally maintain a hashmap that maps vertices to their seed set and a hashmap mapping seeds to their seed cluster weight. Here, logical timestamps will be crucial to efficiently update these terms and the hashmap. By decoupling $w(C(x^*))$ from $h^b(S, C)$, we avoid the large number of sampling tree updates that would have been required when each seed is added to $C$ without the decoupling.

**Maintaining $h^b(S, C)$ And $h_s(S, C)$.** After Algorithm 3 selects $x^*$ in line 3, all the $h^b(S, C)$ terms will initially be $\sum_{x \in S} \deg(x)$, which we already track to compute $f(S, C)$, and all the $h_s(S, C)$ terms will be zero. Additionally, we maintain a hashmap from $X$ to seeds which starts with every node in $X$ mapped to $x^*$. Again we can use timestamps to avoid having to reset this map across queries. Finally we maintain a hashmap from seeds to seed set weights which initially just contains $x^* \to \sum_{x \in X} w(x)$.

From (3), when we add a seed point $y$ to $C$, the only points which can change seed cluster are $y$ and its neighbours, $\{y\} \cup \{x \in X | x \sim y\}$. Therefore, we need to

update $h^b(x, C)$ and $h_s(x, C)$ for the leaves corresponding to these vertices as well as $h^b(S, C)$ and $h_s(S, C)$ for every internal node along the paths from these leaves to the root. Moreover, let $D$ be the set of seeds which lose at least one point to the new seed set. That is, let $D = \{c \in C | |C(c)| > |(C \cup \{y\})(c)|, c \neq x^*\}$. Since the weight of the corresponding seed sets decrease, for every $c \in D$, for every $x \in C(c)$, we will need to update $h_s(x, C)$, and propagate the difference through the internal nodes along the path to the root. Since the number of points in a seed set other than $C(x^*)$ is at most $d_{\max}$ and each seed has at most $d_{\max}$ neighbours, after adding a seed to $C$, the time to update the $h^b$ and $h_s$ values in the tree is $O(d_{\max}^2 \log n)$. We give pseudocode for our dynamic coreset data structure, Data Structure 2, in Appendix C. It has edge update time of $O(\log n)$ and coreset query time of $O\left(k d_{max}^2 \log(n) + \log(n) \log^\star(n) k^2 \varepsilon^{-4}\right)$ time. Appendix E derives the full running times for this data structure and our dynamic coreset spectral clustering data structure, Data Structure 1. The factor of $d_{max}^2 k$ in our query time is very conservative. It assumes that all the neighbours of all the seeds that we sample belong to different seed sets. For many graphs, this is likely impossible.

## 5. Experiments

We perform extensive experiments on a system with an AMD Ryzen 9 7950X 16-Core Processor and 128GB of DDR5 4800MHz RAM. We compare our dynamic CSC data structure against the following baselines: 1) Naive spectral clustering, 2) the static CSC algorithm (Jourdan et al., 2025), 3) LS24 (Laenen & Sun, 2024), 4) Merge&Reduce (Henzinger & Kale, 2020). Following the experiments of Laenen & Sun (2024), we compare these algorithms on real-world and synthetic workloads. Every workload consists of a number of batches of edge insertions or deletions, each followed by a query to get the current partition of the full graph. The details of each workload can be found in Appendix D. In the interest of fairness, we disable any internal buffering so that each edge insertion/deletion is handled one at a time. We record the quality of each query with ARI (Rand, 1971), the time taken to process each batch, and the time taken to process the query and label the current graph. Following Jiang et al. (2024), we use a single round of importance sampling for the coreset algorithms (ours, static CSC, and Merge&Reduce). The full details of each data structure's parameters can be found in the supplementary material.

**Discussion** Figure 1 shows the results on our workloads. Data structures were omitted from experiments if runs took longer than 20,000 seconds. Figure 2 shows a running time comparison between static CSC and ours. All data structures achieve similar ARIs but the edge update and query times vary wildly. With respect to edge update time, LS24 and Merge&Reduce were orders of magnitude slower com-

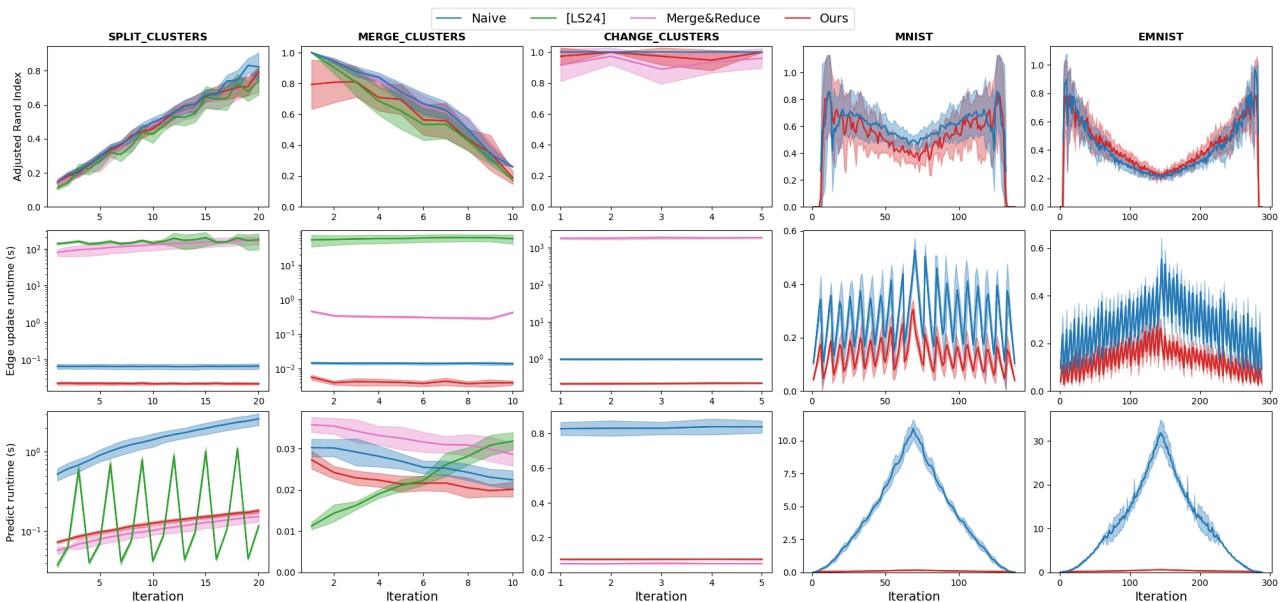

Figure 1. Results on mixed workloads. Shaded regions show standard deviation over 10 runs. Log scales are used when values cross multiple magnitudes.

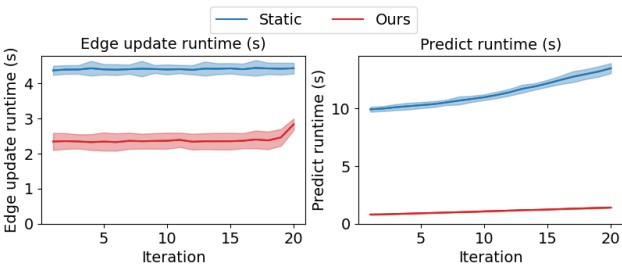

Figure 2. Runtime comparison between the static CSC algorithm and ours on an instance of SPLIT_CLUSTERS with $k = 50$ and $n = 2500$. Sampling from the same distribution, our dynamic data structure achieves the same ARI as the static algorithm. Shaded regions show standard deviation over 10 runs.

pared to the other data structures, and could not be run on the larger experiments without taking longer than 20,000 seconds. With respect to query time, our data structure was orders of magnitude faster than the Naive data structure on larger experiments such as EMNIST while being competitive with LS24 and Merge&Reduce on the smaller experiments. In the comparison against the static algorithm, the query time of the static algorithm increases at a much greater rate compared to our data structure as the experiment progresses, taking ten times longer per query by the end.

## 6. Topology Free Updates With Graph Oracles

We present a version of the dynamic graph clustering problem where edge updates arrive sequentially and our data structure implicitly maintains the topology of the underlying graph using $K_G$ ( Definition 3). Implementations of our data structure either need to maintain their own copy

of the graph, or be given oracle access to it, for example via a graph database driver. For very large graphs, which have to be stored in graph databases, the former is likely impractical. Consider the oracle access setting where our data structure does not store the topology of the graph itself. Our data structure can process edge updates without requiring information about the topology of the graph, only degree information. Indeed, only weighted and unweighted degree information is modified by updates. Conversely, queries do require access to the topology of the graph to sample and extract the coreset graph. This suggests that for very large graphs, instead of consuming a sequence of edge updates and maintaining the graph itself, our data structure could instead consume a sequence of degree updates and be given oracle access to the graph. This also reduces the space required by our data structure to $O(n)$.

## Impact Statement

This paper presents work whose goal is to advance the field of Machine Learning. There are many potential societal consequences of our work, none which we feel must be specifically highlighted here.

## Acknowledgements

Gregory Schwartzman was supported by the following research grants: KAKENHI 21K17703, KAKENHI 25K00370, JST ASPIRE JPMJAP2302, and JST CRONOS JPMJCS24K2.

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

## A. Full CSC Algorithm

In this section we give the full details of the algorithm of (Jourdan et al., 2025). We start with some definitions and notations. Given a weighted graph $G = (V, E, w)$, the conductance of a set $S$ of vertices is defined as $\Phi_G(S) \triangleq w(E(S, V \setminus S))/\text{vol}(S)$ where $w(E(S, V \setminus S))$ is the total weight of edges crossing the cut between $S$ and $V \setminus S$ and $\text{vol}(S)$ is the total weight of edges incident to $S$. We define a $k$-partition of $X$ to be a collection of sets $\Pi = \{\pi_j\}_{j=1}^k$ such that each element of $X$ appears in exactly one member of $\Pi$.

**Definition 5** (centroids). *Given a $k$-partition $\Pi = \{\pi_j\}_{j=1}^k$ of a set $X$, a map $\phi : X \rightarrow \mathcal{H}$ for some Hilbert space $\mathcal{H}$, and a weight function $w : X \rightarrow \mathbb{R}_+$, define the set of centroids of $\Pi$ as $c_w^\phi(\Pi) \triangleq \{c_w^\phi(\pi_j)\}_{j=1}^k$ where $c_w^\phi(\pi_j) = \left(\sum_{x \in \pi_j} w(x)\phi(x)\right) / \left(\sum_{x \in \pi_j} w(x)\right)$.*

**Definition 6** (Normalised cut objective). *Given a graph $G = (V, E)$, the normalised cut problem is to minimise the average conductance over all $k$-partitions of the vertices:* $\min\limits_{\Pi = \{\pi_1, \ldots, \pi_k\}} \text{NC}(G, \Pi)$, *where* $\text{NC}(G, \Pi) = \frac{1}{k} \sum_{j=1}^k \Phi_G(\pi_j)$.

Recall that the normalised cut problem and the kernel $k$-means problem can both be written as the following trace optimisation problems up to a constant (Dhillon et al., 2004):

---

**Normalised Cut**

$$\begin{aligned}
\min \quad & \text{Tr}(D^{-1}A) - \text{Tr}(Z^T D^{-\frac{1}{2}} A D^{-\frac{1}{2}} Z) \\
\text{s.t.} \quad & \mathcal{X} \in \{0,1\}^{n \times k}, \\
& \mathcal{X} 1_k = 1_n, \\
& Z = D^{\frac{1}{2}} \mathcal{X}(\mathcal{X}^T D \mathcal{X})^{-\frac{1}{2}}
\end{aligned}$$

**Weighted Kernel $k$-means**

$$\begin{aligned}
\min \quad & \text{Tr}(WK) - \text{Tr}(Y^T W^{\frac{1}{2}} K W^{\frac{1}{2}} Y) \\
\text{s.t.} \quad & \mathcal{X} \in \{0,1\}^{n \times k}, \\
& \mathcal{X} 1_k = 1_n, \\
& Y = W^{\frac{1}{2}} \mathcal{X}(\mathcal{X}^T W \mathcal{X})^{-\frac{1}{2}}
\end{aligned} \tag{16}$$

---

The full Coreset Spectral Clustering algorithm is given in Algorithm 4. Followed by an intuitive explanation in Figure 3.

---

**Algorithm 4** CORESET SPECTRAL CLUSTERING

---

1: **Input:** Graph $G = (V, E)$, $k$, adjacency $A_G$, degree $D_G$
2: $K_G, W_G \leftarrow D_G^{-1} A_G D_G^{-1}, \ D_G$
3: $V', W_H \leftarrow$ an $\varepsilon$-coreset for kernel $k$-means on $(V, K_G, W_G)$
4: $A_H \leftarrow W_H K_G(V') W_H$          $\triangleright K_G(V')$ is the principal submatrix of $K_G$ indexed by $V'$.
5: $\Pi \leftarrow \text{SPECTRALCLUSTERING}(A_H, k)$          $\triangleright \Pi = \{\pi_j\}_{j=1}^k$ is a $k$-partition of $V'$.
6: $\Pi' \leftarrow$ partition of $V$ assigning each $x \in V$ to the closest centroid in $c_{w_H}^\phi(\Pi)$
7: **return** $\Pi'$

---

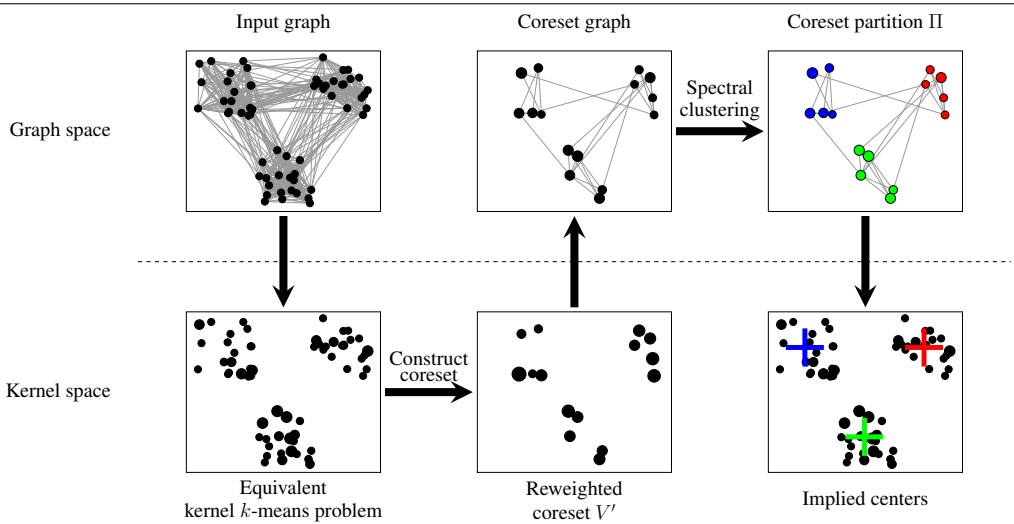

*Figure 3.* Sketch of the Coreset Spectral Clustering Algorithm, taken from (Jourdan et al., 2025).

Now let us state formally the results of (Jourdan et al., 2025).

**Theorem 3.** *Given a graph $G = (V, E)$ and an $\alpha$-approximation algorithm for the normalised cut problem with $k$ clusters (16), Algorithm 4 returns a $k$-partition of $V$ that is a $\frac{1+\varepsilon}{1-\varepsilon}\alpha$-approximation to the optimal normalised cut value for $G$. The running time of Algorithm 4 is the sum of the running time of the $\varepsilon$-coreset algorithm, SPECTRALCLUSTERING, and labelling $V$.*

### A.1. Full Dynamic CSC Data Structure

Here we list the pseudocode for our dynamic coreset spectral clustering data structure, that uses our dynamic coreset data structure, Data Structure 2, as a building block.

---

**Data Structure 1: DYNAMIC CSC Data Structure**

1: **Fields:**
2:     `coreset`: an instance of Data Structure 2
3:     `alg`: an $\alpha$-approx. algorithm for Normalised Cut (input: adjacency matrix, $k$)
4:
5: **Update**($\text{self}, \text{mode}, u, v, w$)
6: **if** mode == `"insert"` **then**
7:     self.coreset.**InsertEdge**($u, v, w$)
8: **else**
9:     self.coreset.**DeleteEdge**($u, v, w$)
10: **end if**
11:
12: **Query**($\text{self}, k, \varepsilon, \sigma, Y$)
13:     $G = (V, E)$, $D_G$, $K_G = D_G^{-1} A_G D_G^{-1} + \sigma D_G^{-1}$, $W_G = D_G$                    ▷ Implicitly maintained
14:     $V', W_H \leftarrow$ self.coreset.**ExtractCoreset**($k, \sigma, \varepsilon$)       ▷ $\varepsilon$-coreset for kernel $k$-means on $(V, K_G, W_G)$.
15:     $A_H \leftarrow W_H K_G(V') W_H$
16:     $\Pi \leftarrow$ self.**alg**($A_H, k$)
17:     $\Pi' \leftarrow$ partition assigning each $x \in Y$ to the closest centroid in $c_{w_H}^\phi(\Pi)$
18:     **return** $\Pi'$

---

### A.2. Labelling $Y$

To compute $\Pi'$, line 17 of Data Structure 1, we must fist compute the self inner products of the centroids in $c_{w_H}^\phi(\Pi)$. That is, for every $c_w^\phi(\pi_j)$ in $c_w^\phi(\Pi)$, we need to compute $\langle c_w^\phi(\pi_j), c_w^\phi(\pi_j) \rangle$. Since the coreset graph has at most $O(k^2\varepsilon^{-4})$ vertices, the largest component of $\Pi$ may contain $O(k^2\varepsilon^{-4})$ vertices. Thus the time to compute the self inner products takes at most $O(k^4\varepsilon^{-8})$ time.

Then for each $x \in Y$, we must find the closest centroid. Due to the nature of the graph kernel, it suffices to check which centroids $x$'s neighbours contribute to. In particular, we only need to check neighbours which are in the coreset graph. This can be achieved in two ways: 1) Iterate over the neighbours of $x$ and check if they are in the coreset graph, or 2) For each vertex $y$ in the coreset graph, check if $y$ is a neighbour of $x$. Using the first approach to label $Y$ takes $O(\text{vol}(Y))$ time and the second approach takes $|Y| k^2\varepsilon^{-4}$ time. Thus the total time to label $Y$ is $O(k^4\varepsilon^{-8} + \min\{|Y| k^2\varepsilon^{-4}, \text{vol}(Y)\})$.

## B. Sampling Trees

Suppose we have a set of $n$ objects $X$ with associated probability distribution $p(x) = \frac{l(x)}{\sum_{x \in X} l(x)}$ for all $x \in X$, where $l : X \to \mathbb{R}_{\geq 0}$. Further define $l(S) = \sum_{x \in S} l(x)$ for all $S \subseteq X$. We refer to $l(x)$ as the contribution of $x$ and $l(S)$ as the contribution of $S$. Sampling trees provide an efficient way to sample from $p(x)$, and an efficient way to update the contribution of a point (Wong & Easton, 1980). Suppose we have nested sets $S_1 \subset S_2 \subset S_3 \subset \cdots \subset S_m \subseteq X$. For any $x \in S_1$, we have that

$$\Pr\left[x \text{ is sampled}\right] = \frac{l(x)}{l(X)} = \frac{l(S_m)}{l(X)} \frac{l(S_{m-1})}{l(S_m)} \cdots \frac{l(x)}{l(S_1)}. \tag{17}$$

Following this decomposition, sampling trees maintain the contribution of points in $X$ as leaves and internal nodes maintain the contribution corresponding to the sum of their children, in the form of a balanced binary tree.

Let $T$ be a sampling tree for sampling elements of $X$ from distribution $p(x)$. To sample a data point according to $p(x)$, we start at the root node of $T$ and recursively sample a child node with probability equal to the child's contribution divided by

the parent's contribution until we reach a leaf, and return the element of $x$ stored there. From equation 17, this is equivalent to sampling points according to $p(x)$ while taking time $O(\log n)$.

To update the contribution of a point, the contribution at the leaf is overwritten and the difference is subtracted from every internal node in the path from the target leaf node to the root. Alternatively, the contribution of the leaf is overwritten and every internal node in the path to the root recomputes its contribution as the sum of its children. Both approaches take $O(\log n)$ time with the former being faster and the latter being more numerically stable. As such, sampling and updating a contribution takes $O(\log n)$ time while building a sampling tree from scratch takes time $O(n)$.

## C. Our Data Structure

In this Section we give the pseudocode for our data structure, Data Structure 2.

---

**Data Structure 2: DYNAMIC CSC Data Structure**

1: **Fields:**
2:   $\texttt{T}$: global logical timestamp                                           ▷ Used to invalidate stale query data.
3:   $\texttt{G}$: dynamic graph $G = (V, E, w)$ as hashmap-of-hashmaps
4:   $\texttt{MaxDeg}$: max-heap of node degrees
5:   $\texttt{Tree}$: sampling tree with vertices as leaves; each node stores:
6:     $\texttt{Node.T}$: logical timestamp
7:     $\texttt{Node.vol}$: $\sum_{x \in S} \deg(x)$                                  ▷ $S$ is the vertex set represented by the node.
8:     $\texttt{Node.size}$: $|S|$
9:     $\texttt{Node.}f_\delta$: $\sum_{x \in S} f_\delta(x, C)$                      ▷ Reset to 0 if $\texttt{Node.T}$ is stale.
10:     $\texttt{Node.}h^b$: $\sum_{x \in S} h^b(x)$                                  ▷ Reset to $\texttt{Node.vol}$ if stale.
11:     $\texttt{Node.}h_s$: $\sum_{x \in S} h_s(x)$                                  ▷ Reset to 0 if stale.
12:   $\texttt{SeedMap}$: map $x \mapsto (\texttt{t}, \texttt{seed})$
13:     $\texttt{t}$: logical timestamp
14:     $\texttt{seed}$: seed vertex                                                  ▷ Reset to current $x^*$ if $\texttt{t}$ is stale.

15:
16: **Init**($\texttt{self}$)
17:   $\texttt{self.T} \leftarrow 0$
18:   $\texttt{self.G} \leftarrow$ empty graph
19:   $\texttt{self.MaxDeg} \leftarrow$ empty max-heap
20:   $\texttt{self.Tree} \leftarrow$ empty sampling tree
21:   $\texttt{self.SeedMap} \leftarrow$ empty hashmap

22:
23: **InsertEdge**($\texttt{self}, u, v, w$)
24:   Insert edge $\{u, v\}$ with weight $w$ into $\texttt{self.G}$
25:   Increment degrees of $u$ and $v$ in $\texttt{self.MaxDeg}$
26:   Ensure $u, v$ exist as leaves in $\texttt{self.Tree}$; promote as required        ▷ All node fields default to 0.
27:   Update leaf fields for $u$ and $v$ to reflect the insertion
28:   Propagate field deltas up to the root of $\texttt{self.Tree}$

29:
30: **DeleteEdge**($\texttt{self}, u, v, w$)
31:   Delete edge $\{u, v\}$ with weight $w$ from $\texttt{self.G}$
32:   Decrement degrees of $u$ and $v$ in $\texttt{self.MaxDeg}$
33:   Update leaf fields for $u$ and $v$ to reflect the deletion
34:   Propagate field deltas up to the root of $\texttt{self.Tree}$
35:   If $u$ becomes disconnected: swap its leaf with last leaf, delete it,
36:     then propagate; demote internal nodes to leaves as needed. Repeat for $v$.

37:
38: **F**($\texttt{Node}, \sigma, x^*, T$)                                           ▷ From (5).
39: **if** $\texttt{Node.T} < T$ **then**
40:     $\texttt{Node.T} \leftarrow T$
41:     $\texttt{Node.}f_\delta \leftarrow 0$

---

42:    **end if**
43:       **return** `Node.size` $\cdot \sigma + \frac{\sigma}{\deg(x^*)} \cdot$ `Node.vol` $-$ `Node.`$f_\delta$
44:
45: **G**(`self`, `Node`, $\sigma, x^*, w_{x^*}$)
46:     $f_S \leftarrow$ `self.F(Node,` $\sigma, x^*,$ `self.T)`
47:     $f_X \leftarrow$ `self.F(self.Tree.root,` $\sigma, x^*,$ `self.T)`
48: **if** `Node.T` $<$ `self.T` **then**
49:      `Node.T` $\leftarrow$ `self.T`
50:      `Node.`$h^b \leftarrow$ `Node.vol`
51:      `Node.`$h_s \leftarrow 0$
52: **end if**
53:     **return** $\frac{f_S}{f_X} + \frac{\texttt{Node.}h^b}{w_{x^*}} +$ `Node.`$h_s$                           $\triangleright$ $w_{x^*}$ corresponds to $w(C(x^*))$; from (15).
54:
55: **ExtractCoreset**(`self`, $k, \sigma, \varepsilon$)
56:     `self.T` $\leftarrow$ `self.T` $+1$
57:     $x^* \leftarrow$ argmax degree from `self.MaxDeg`
58:     `SeedWeight` $\leftarrow \{\, x^* \mapsto$ `self.Tree.root.vol` $\}$      $\triangleright$ SeedWeight maps seeds to seed-weights (degrees).
59:     `self.Repair(`$x^*, \sigma, x^*,$ `SeedWeight)`
60:     Draw $x$ uniformly at random from $V$
61:     `self.Repair(`$x, \sigma, x^*,$ `SeedWeight)`
62: **for** $i = 1, \ldots, k-1$ **do**
63:      Sample $x$ from `self.Tree` using `self.F` to compute contributions
64:      `self.Repair(`$x, \sigma, x^*,$ `SeedWeight)`
65: **end for**
66:     $D \leftarrow \emptyset$
67:     $N \leftarrow O\left( \frac{k^2 \log^2(k) \log(\|V\|)}{\varepsilon^4} \right)$                 $\triangleright$ $\|V\|$ is the number of nodes in the graph.
68: **for** $j = 1, \ldots, N$ **do**
69:      $(x, p) \leftarrow$ sample and its probability from `self.Tree`
70:      $D \leftarrow D \cup \{(x, \deg(x)/(pN))\}$
71: **end for**
72:     $X_1 \leftarrow$ weighted dataset $D$         $\triangleright$ In practice we stop after a single round of importance sampling.
73:     $i \leftarrow 1$
74: **repeat**
75:      $i \leftarrow i + 1$ and $\varepsilon_i \leftarrow \varepsilon / (\log^{(i)} \|X_1\|_0)^{1/4}$
76:      $X_i \leftarrow$ IMPORTANCE-SAMPLING$(X_{i-1}, \varepsilon_i)$            $\triangleright$ Alg. 2
77: **until** $\|X_i\|_0$ does not decrease compared to $\|X_{i-1}\|_0$
78:     **return** $X_i$
79:
80: **Repair**(`self`, $x, \sigma, x^*,$ `SeedWeight`)
81:     $L_x \leftarrow$ leaf in `self.Tree` for vertex $x$             $\triangleright$ Maintain maps between graph vertices and tree leaves.
82:     $w_x \leftarrow \deg(x)$
83:     `OldSeed` $\leftarrow$ `self.SeedMap[`$x$`]`                    $\triangleright$ Overwrite to $x^*$ if stale.
84:     `SeedWeight[OldSeed]` $\leftarrow$ `SeedWeight[OldSeed]` $- w_x$
85:     `SeedWeight[`$x$`]` $\leftarrow w_x$
86:     $f' \leftarrow L_x.f_\delta$
87:     $L_x.f_\delta \leftarrow \sigma + \deg(x)\left( \frac{\sigma}{\deg(x^*)} \right)$            $\triangleright$ Sets $f(x, C)$ to 0 via $f_\delta \leftarrow f^b$; see (5).
88:     Add $(L_x.f_\delta - f')$ to $L.f_\delta$ along path $L_x \rightsquigarrow$ root
89:     `self.SeedMap[`$x$`]` $\leftarrow x$                         $\triangleright$ Also update timestamp.
90:     `OldSeeds` $\leftarrow \{$`OldSeed`$\}$
91:     $S_x \leftarrow \{\, y \sim x \mid w(y)\Delta(y, x) <$ `self.F(`$L_y, \sigma, x^*,$ `self.T)` $\}$
92: **for** $z \in S_x$ **do**
93:      $L_z \leftarrow$ leaf in `self.Tree` for $z$

```
94:         f' ← L_z.f_δ
95:         L_z.f_δ ← σ + deg(z)( σ/deg(x*) ) − Δ(z, x)                          ▷ From (6).
96:         Add (L_z.f_δ − f') to L.f_δ along path L_z ⤳ root
97:         w_z ← deg(z)
98:         OldSeed ← self.SeedMap[z]                                    ▷ Overwrite to x* if stale.
99:         OldSeeds ← OldSeeds ∪ {OldSeed}
100:        SeedWeight[OldSeed] ← SeedWeight[OldSeed] − w_z
101:        SeedWeight[x] ← SeedWeight[x] + w_z
102:        self.SeedMap[z] ← x                                          ▷ Also update timestamp.
103:    end for
104:    S'_x ← {x} ∪ S_x
105:    for z ∈ S'_x do
106:        L_z ← leaf in self.Tree for z
107:        h'_b ← L_z.h^b                                               ▷ Reset to L_z.vol if stale.
108:        h'_s ← L_z.h_s                                               ▷ Reset to 0 if stale.
109:        L_z.h^b ← 0
110:        L_z.h_s ← deg(z)/SeedWeight[x]                               ▷ Case 2 of (12).
111:        Subtract h'_b from L.h^b along path L_z ⤳ root
112:        Add (L_z.h_s − h'_s) to L.h_s along path L_z ⤳ root
113:    end for
114:    for s ∈ OldSeeds \ {x*} do
115:        (Update h_s for vertices assigned to s)             ▷ Seed-sets that shrank, excluding C(x*).
116:        for z ∈ { y ∼ s | self.SeedMap[y] = s } do
117:            L_z ← leaf in self.Tree for z
118:            h'_s ← L_z.h_s                                          ▷ Should never be stale.
119:            L_z.h_s ← deg(z)/SeedWeight[s]
120:            Add (L_z.h_s − h'_s) to L.h_s along path L_z ⤳ root
121:        end for
122:    end for
```

## D. Experiment Workloads

Our experiments are dividing into real-world and synthetic dynamic graph workloads, which are fully described in this section.

**Real-World Workloads.** For our real-world workloads, we consider the MNIST and EMNIST datasets (Lecun et al., 1998; Cohen et al., 2017). For each dataset we define a workload based on the $k$-nearest neighbours graph, with $k = 300$. We begin with an empty graph, and then the nodes (and associated edges) in the $k$-nearest neighbour graph are added in batches of 1000. The nodes are inserted in order of cluster id, and the order within clusters is randomized. After every node has been inserted, they are removed again in the reverse order to which they were inserted.

**Synthetic Workloads.** Our synthetic workloads are all based on the stochastic block model (SBM) for generating random graphs. We use two insertion-only workloads which were originally defined by Laenen and Sun (Laenen & Sun, 2024), and a new fully-dynamic workload.

In the SPLIT_CLUSTERS workload, we initially add a graph drawn from the standard SBM with the number of clusters set to $k = 30$, and with $n = 300$ vertices in each cluster. We include each edge inside a cluster with probability $p = 0.5$ and we include edges between clusters with probability $q = (nk)^{-1}$. Then, we perform 10 phases in which we select a subset of 300 vertices in the graph, and add a clique on those vertices so that they effectively become a new cluster. We ensure that sets of vertices chosen at each update are disjoint. After each phase, the cluster structure is queried from the dynamic data structure.

In the MERGE_CLUSTERS workload, we again begin with a graph drawn from the standard SBM. We set the number of clusters to be 20, each containing 100 vertices. We set the SBM parameters to be $p = 0.5$, and $q = (nk)^{-1}$. We then perform 10 phases in which we select two clusters, and add each edge connecting those clusters with probability 0.95, effectively merging those two clusters into one larger cluster. After each phase, the cluster structure is queried from the dynamic data structure.

The third synthetic workload, referred to as CHANGE_CLUSTERS is fully-dynamic. We begin with a graph drawn from the

standard SBM, with 10 clusters each containing 1000 nodes. The parameters of the SBM are $p = 0.5$ and $q = 10^{-4}$. We proceed in 5 phases. In each phase, two clusters $C_1$ and $C_2$ of the currently maintained graph are chosen, and every edge incident to these clusters is removed. Then, $C_1$ and $C_2$ are each split in half into vertex sets $\{C_{1,1}, C_{1,2}\}$ and $\{C_{2,1}, C_{2,2}\}$ and we create two new clusters: $C_1' = C_{1,1} \cup C_{2,1}$ and $C_2' = C_{1,2} \cup C_{2,2}$, adding internal and outgoing edges with probabilities $p$ and $q$ respectively. After each phase, the cluster structure is queried from the dynamic data structure.

## E. Running Time Calculations new section

In this Appendix, we derive the running times given in Table 1.

**Theorem 4** (Theorem 3.1 (Jiang et al., 2024))**.** *Given an $n$-point weighted dataset $X$, oracle access to a kernel $K$ : $X \times X \to \mathbb{R}$, $0 < \varepsilon < 1$, and integer $k \geq 1$, Algorithm 1 constructs a reweighted subset $S \subseteq X$ of size*

$$\|S\|_0 = O(\varepsilon^{-4} k^2),$$

*that with high constant probability [4] is an $\varepsilon$-coreset for kernel $(k, 2)$-CLUSTERING with respect to $X$ and $K$. The running time is $O(nk + \log(n) \log^\star(n) k^2 \varepsilon^{-4})$.*

**Remark 1.** *The factor of $nk$ in the running time of Theorem 4 is due to the use of naive $D^2$-sampling in Algorithm 2. In our data structure, the corresponding running time is only $O(kd_{max}^2 \log(n))$ ( Section 4). Thus, the running time of* **ExtractCoreset** *in Data Structure 2 is $O(kd_{max}^2 \log(n) + \log(n) \log^\star(n) k^2 \varepsilon^{-4})$.*

**Remark 2.** *Given the running time of* **ExtractCoreset**, *the time required for Data Structure 1 to perform* **Query**$(k, \varepsilon, \sigma, Y)$ *is $O(kd_{max}^2 \log(n) + \log(n) \log^\star(n) k^2 \varepsilon^{-4} + T_{NC}(\varepsilon^{-4} k^2) + \varepsilon^{-8} k^4 + \min\{|Y| \, k^2 \varepsilon^{-4}, \mathrm{vol}(Y)\})$, where $T_{NC}(n, k')$ is an $\alpha-$approximation algorithm for the normalised cut problem with $k'$ clusters (16) and $\mathrm{vol}(Y)$ is the sum of unweighted degrees of all nodes in $Y$. The final two terms in the running time are the cost of computing the norms of each coreset centroid in kernel space, and then using them to label everything in $Y$; Section A.2 gives the derivation. The running time of* **Update** *is $O(\log(n))$.*

**Theorem 5** ((Henzinger & Kale, 2020))**.** *Assume that there is a static coreset construction algorithm for a problem $P$ with linear cost function that* **a)** *runs in time $t_P(n_s, \varepsilon_s, \lambda_s)$,* **b)** *always outputs a set of cardinality at most $s_P(\varepsilon_s, \lambda_s)$, and* **c)** *has the guarantee that the output is an $\varepsilon_s$-coreset with probability at least $1 - \lambda_s$, where $n_s$ is the number of input points. Then there is a fully-dynamic coreset-construction algorithm for $P$ that,* **a)** *always maintains an output set of cardinality at most $s_P(\varepsilon, \lambda)$,* **b)** *has the guarantee that the output is an $\varepsilon$-coreset with probability at least $1 - \lambda$, and* **c)** *has worst-case update time*

$$O\Big((\log n) \cdot t_P\big(s_P^*, \tfrac{\varepsilon}{\log n}, \tfrac{\lambda}{n}\big)\Big),$$

*where $s_P^* = s_P\big(\tfrac{\varepsilon}{\log n}, \tfrac{\lambda}{n}\big)$, and $n$ is the current number of points.*

**Remark 3.** *Using the algorithm of (Jiang et al., 2024) (Theorem 4) as the static coreset algorithm for Theorem 5, with high constant probability, Merge&Reduce maintains a coreset of size $O(\varepsilon^{-4} k^2)$ after every update. To compute the update time, notice that $s_P^\star = O(\varepsilon^{-4} k^2 \log^4(n))$ and so the update time is*

$$O\Big( \log(n) \cdot \big(\varepsilon^{-4} k^2 \log^4(n) k + \log^\star(\varepsilon^{-4} k^2 \log^4(n)) \cdot (k^2 \log^2(k) \varepsilon^{-4} \log^4(n))\big)\Big)$$

$$= O\Big(\varepsilon^{-4} k^3 \log^5(n)\Big)$$

**Remark 4.** *Given Merge&Reduce maintains a coreset of size $O(\varepsilon^{-4} k^2)$, the query time of merge & reduce is $O\Big(T_{NC}(\varepsilon^{-4} k^2, k) + \varepsilon^{-8} k^4 + \min\{|Y| \, k^2 \varepsilon^{-4}, \mathrm{vol}(Y)\}\Big)$.*

## F. Formal Statement Of Main Results

As our data structure supports set queries, let us define what does it mean for a subset of the nodes to be properly labelled with respect to the normalised cut objective.

**Definition 7.** *($\alpha$-consistent labelling) Given a graph $G(V, E)$, we say that a labelling $c$ for $Y \subseteq V$ is $\alpha$-consistent for the normalised cut problem if there exists a labelling $c'$ for $V$ that agrees with $c$ on $Y$ and $c'$ corresponds to and $\alpha$-approximation to the normalised cut objective for $V$.*

---

[4]That is, for any constant $\lambda$ the algorithm succeeds with probability $1 - \lambda$.

Before we state the formal version of Theorem 2. Recall that a dynamic weighted graph $\mathcal{G} = G_1(V_1, E_1), G_2(V_2, E_2), \ldots$ is either a finite or infinite sequence of graphs where $G_i, G_{i+1}$ only differ by at most a single edge corresponding to an edge insertion or deletion. We assume that when all edges incident to a node are removed, that node is removed. When $(u, v)$ is added at step $i$, we have $u, v \in V_i$. Let $n_i = |V_i|, d_{max}^i = \max_{v \in V_i}\{d(v)\}$.

**Theorem 6** (Main result (Formal))**.** *Given a dynamic weighted graph $\mathcal{G}$ and an $\alpha$-approximation algorithm for the normalised cut problem with $k'$ clusters (16) running in time $T_{NC}(n, k')$, upon the $i$-th change, Data Structure 2 can be updated in time $O(\log n_i)$. When queried with $Y \subseteq V_i, k \in [n_i], \varepsilon \in (0, 1)$ between the $i$-th and $(i + 1)$-th change it runs in time $O(k(d_{max}^i)^2 \log(n_i) + \log(n_i) \log^\star(n_i)k^2\varepsilon^{-4} + T_{NC}(\varepsilon^{-4}k^2) + \varepsilon^{-8}k^4 + \min\{|Y| k^2\varepsilon^{-4}, \mathrm{vol}(Y)\})$ and returns a labelling of $Y$ that is $\alpha(1 + \varepsilon)/(1 - \varepsilon)$-consistent for the normalised cut problem, with high constant probability.*

