# OpenReview forum: "Fully Dynamic Coreset Spectral Clustering"
_ICML.cc/2026/Conference — ICML 2026 regular_

### Official Review · Reviewer_n4y4 · 2026-03-12

**Soundness:** 3
**Presentation:** 3
**Significance:** 3
**Originality:** 3
**Overall Recommendation:** 4
**Confidence:** 4

**Summary:**

1. This paper studies the problem of spectral clustering in a fully dynamic manner. Given a dynamic weighted graph, the authors present a data structure supporting edge/node updates and cluster membership queries for the Normalised Cut problem.
2. A novel Just-in-Time Sampling Tree framework is introduced to dynamically maintain graph kernel coresets, enabling worst-case O(log n) edge update time and sublinear query time while preserving an given approximation ratio.
3. Experiments are conducted to show the effectiveness and efficiency of the proposed data structure on real-world datasets, significantly outperforming several state-of-the-art methods.

**Compliance With Llm Reviewing Policy:**

Affirmed.

**Final Justification:**

Thank you for the clarification. It resolves my main concerns, so I have raised my score.

**Key Questions For Authors:**

1. The experimental results do not seem to fully match the complexity comparison shown in Table 1. According to Table 1, the update time of the Naive and Static methods is O(1), yet in the experiments their running time is higher than the proposed method with O(log n) complexity. It would be helpful if the paper could provide a discussion for this discrepancy.
2. The layout of the paper could be further optimized. For example, reducing excessive white space, improving the placement of figures and tables, and adding a Conclusion section would significantly improve the overall readability.
3. The references are somewhat limited.

**Limitations:**

yes

**Strengths And Weaknesses:**

S1. The paper is well-motivated, addressing the problem of maintaining spectral clustering in fully dynamic environments.

S2. The proposed Just-in-Time Sampling Trees method is novel, providing a worst-case O(log n) update time with theoretical garantees.

S3. The theoretical analysis is sound. The authors provide solid guarantees for both the approximation ratios and the computational complexity.

W1. The experimental evaluation is not comprehensive enough as it mainly compares performance under different workloads, neglecting the impact of various graph types and scales. For instance, according to Table 1, if d_max^2 > n, the proposed method might actually be less efficient than static approaches in both update and query scenarios, which needs more discussion.

W2. The exact magnitude of T_NC(n, k) is not very clear. It seems that the value of T_NC should decrease as the first parameter n becomes smaller. However, the paper does not explicity specify the value of \epsilon in experiments, and the value of n also does not appear to be very large (n < 10^5) according to Appendix D. In this case, it is confusing which term is larger in practice, T_NC(n, k) or T_NC(\epsilon^-4 k^2, k), and how big the difference is. Some additional clarification would be helpful.

W3. The Conclusion section is missing.

---

> ### Author Rebuttal · Authors · 2026-03-27
>
> Thank you for your comments and questions. We think this work is an exciting contribution to the area of dynamic graph algorithms and hope our response below will convince you of its merits.
>
> **Weaknesses**
>
> W1: The experimental evaluation is not comprehensive enough as it mainly compares performance under different workloads, neglecting the impact of various graph types and scales. For instance, according to Table 1, if $d_{max}^2 > n$, the proposed method might actually be less efficient than static approaches in both update and query scenarios, which needs more discussion.
>
> In our practical experiments, it is actually the case that $d_{\max}^2 > n$, and we find that our approach out-performs the others. This illustrates the point made to other reviewers that while a dependence on $d_{\max}^2$ shows up in the theoretical analysis, the maximum degree does not appear to have a significant impact in practice. If you can think if another graph type and scale to try we will add it to our experiments.
>
> We must note that even on the ones we tried, the edge update times of the LS24 and Merge\&Reduce algorithms become prohibitive, for example on MNIST and EMNIST we gave up after each run took more than 20,000 seconds. We will add a discussion of the dependence on $d_{\max}^2$. As we have mentioned to other reviewers, this is a conservative bound that might be improved in future work.
>
>
> W2: The exact magnitude of $T_{NC}(n, k)$ is not very clear. It seems that the value of $T_{NC}$ should decrease as the first parameter n becomes smaller. However, the paper does not explicity specify the value of $\epsilon$ in experiments, and the value of n also does not appear to be very large ($n < 10^5$) according to Appendix D. In this case, it is confusing which term is larger in practice, $T_{NC}(n, k)$ or $T_{NC}(\epsilon^-4 k^2, k)$, and how big the difference is. Some additional clarification would be helpful.
>
> We think there might be some confusion over the meaning of $T_{NC}(n, k)$ and the relationship between $\epsilon$ and coreset size.
>
> In the abstract, we define it to be the running time of an $\alpha$-approximation algorithm for the normalised cut problem, when the input graph has $n$ vertices and $k$ clusters. When we write $T_{NC}(\epsilon^{-4} k^2, k)$, we are referring to the running time of the normalized cut algorithm on a graph with $\epsilon^{-4} k^2$ nodes and $k$ clusters. This corresponds to the step where we run our NC algorithm on the coreset graphs.
>
> The value of $\epsilon$ in the experiments can be inferred from the coreset sizes we use in our experiments. In theory, the corset size is dictated by $\epsilon$ via line 4 of Algorithm 2 (the $||X||_0$ disappears after $\log^\star(n)$ rounds of importance sampling). However, even for coarse values of $\epsilon$, this is far far larger than what is needed in practice, so we typically use much smaller coreset sizes. We will make this clear in the final version of our paper.
>
> Even for the relatively small graphs that we test on, we can see a massive difference between $T_{NC}(n, k)$ and $T_{NC}(\epsilon^{-4} k^2, k)$. Indeed, the bottom right plot in Figure 1 clearly shows that the naive algorithm's query time ($T_{NC}(n, k)$) is much slower than ours. In theory, in the extreme case where $k$ and $\epsilon$ are constant, $T_{NC}(\epsilon^{-4} k^2, k)=O(1)$ because the normalised cut problem can be solved exhaustively in constant time when the input graph is constant sized.
>
> **Questions**
>
> Q1: According to Table 1, the update time of the Naive and Static methods is O(1), yet in the experiments their running time is higher than the proposed method with O(log n) complexity. It would be helpful if the paper could provide a discussion for this discrepancy.
>
> The differences in update time are due to the fact that the algorithms are implemented across a range of programming languages, and for some of our experiments, the number of nodes is constant over time. Due to this, the overall trends are more informative than the exact numbers.
>
> For example, while the edge update time of our algorithm appears to be slightly faster than the naive method on Merge\_Clusters (roughly 2x), the update time of Merge\&Reduce and LS24 are tens to hundreds of times slower.
>
> Q2: The layout of the paper could be further optimized. For example, reducing excessive white space, improving the placement of figures and tables, and adding a Conclusion section would significantly improve the overall readability.
>
> We will address these presentation issues in the final version.
>
> Q3. The references are somewhat limited.
>
> If there are specific references to related work which you feel are missing, then we would be happy to add them with discussion.

---

> > ### Author Rebuttal · Reviewer_n4y4 · 2026-04-03
> >
> > Thank you for the clarification. It resolves my main concerns, so I have raised my score.

---

### Official Review · Reviewer_Br6q · 2026-03-12

**Soundness:** 4
**Presentation:** 3
**Significance:** 3
**Originality:** 3
**Overall Recommendation:** 5
**Confidence:** 3

**Summary:**

The paper develops an algorithm for a dynamic form of
spectral clustering. The algorithm permits edge and node
additions/deletions and querying the cluster IDs for a subset
of the nodes. A good data structure that efficiently supports
the update and query operations is developed. The query time is
an improvement over currently known approaches. The paper
establishes rigorous performance guarantees and includes
experimental results to illustrate the resulting performance
improvement.

**Compliance With Llm Reviewing Policy:**

Affirmed.

**Key Questions For Authors:**

(a) Page 2, left column, Theorem 1: The running time mentioned for the
algorithm doesn't explicitly involve d_{avg} which is defined on lines
64--65. It may be better to mention that vol(G) is given by n \times d_{avg}
to make the dependence of d_{avg} more explicit.

(b) Page 2, left column, the paragraph on "Dynamic Clustering": Since
your model does not allow update of edge weights, it is not clear
whey you say that "G_i and G_{i+1} differ by at most a single edge".
If you don't permit edge weight updates, won't they always differ
by a single edge? (It is possible that this reviewer is missing
something simple here.)

(c) In Table 1, by looking at the query time entries for your algorithm
and that for Merge-Reduce, it is not readily clear why your update time
is always better than that of Merge-Reduce. The reason for this concern is
that your update time depends on (d_{max})^2 while that of Merge-Reduce
depends on Vol(Y), which seems to depend only on d_{max}. Please clarify this.

(d) Page 3, lines 143--144: You mention that "k can be chosen at query time".
This requires clarification since the query only allows a subset Y of
nodes to be specified; it doesn't allow the user to specify the value of k.

(e) Page 4, Definition 3: After specifying the expression for K, you mention
that \sigma is chosen such that K is positive definite. Is such a value of
\sigma unique? If not, is your algorithm sensitive to the chosen
value of \sigma?

Minor comments:

(i) In Definition 2: "equation 1" ---> "Equation (1)".

(ii) The notation "x \tilde y" (to indicate that there is an edge
between nodes x and y) appears for the first time in Definition 3.
It is better to introduce this notation earlier in Section 2.

**Limitations:**

Yes

**Strengths And Weaknesses:**

Strengths:

(a) The problem studied (dynamic version of spectral clustering) is
of considerable interest to the ML community. The paper provides a nice
discussion of prior work.

(b) To this reviewer's knowledge, the new data structure (which is
presented in great detail in the supplement) is novel. The corresponding
update algorithm to obtain an improved query time is nontrivial. The
algorithm will be of interest to practitioners as well.

(c) Overall, the paper is written well. A decent amount of intuitive explanation
is included to point out why previous approaches are inadequate to
achieve the query time obtained in this paper.

Weakness: This reviewer can only identify two (very) minor weaknesses.
First, due to the complicated nature of the running time expressions,
it takes time for a reader to understand the improvement. (This may
not be avoidable since the problem involves many  parameters.)
Second, in a few places, some additional explanation (possibly in
the supplement) may make it easier to follow the discussion. Some of these
are mentioned in the "Key Questions" section.

---

> ### Author Rebuttal · Authors · 2026-03-27
>
> Thank you for your comments and questions.
>
> **Questions**
>
> a) Theorem 1: The running time mentioned for the algorithm doesn't explicitly involve $d_{avg}$
>
> This is a bit of a typo thanks. the last sentence (line 64) should stop at the last comma. Theorem 1 assumes it is given a coreset. The $d_{avg}$ comes from the runtime of the static CSC algorithm.
>
> b) If you don't permit edge weight updates, won't they always differ by a single edge?
>
> We say $G_{i}$ and $G_{i+1}$ differ by at most a single edge to rule out sequences of graphs which differ by more than this. In our analysis, we can simulate edge weight updates with an edge deletion followed by an edge insertion. In practice, updating edge weights is straight-forward since it only requires us to update the degree max-heap and the volume fields of the sampling tree data structure.
>
> c) ...your update time depends on $(d_{max})^2$ while that of Merge-Reduce depends on Vol(Y), which seems to depend only on $d_{max}$. Please clarify this.
>
> Our update time is always better than Merge\&Reduce because we sample coresets at query time, Merge\&Reduce samples coresets after every update. This is also why Merge \& Reduce only pays for the time to run spectral clustering and labelling $Y$ at query time. Note that the $d_{max}$ dependence only appears in the query time of our data-structure.
>
> d) "k can be chosen at query time". This requires clarification since the query only allows a subset Y of nodes to be specified; it doesn't allow the user to specify the value of k.
>
> Our full query algorithm is given on line 12 of Data Structure 1, and supports specifying the value of $k$. If we have suggested otherwise elsewhere in the paper, then we would be happy to correct it.
>
> e)  Is such a value of $\sigma$ unique? If not, is your algorithm sensitive to the chosen value of $\sigma$?
>
> Fortunately, our algorithm is not sensitive to $\sigma$. Indeed, Jourdan et. al. (2025) were the first to notice this with static CSC. We found choosing a value large enough is straight-forward and use $\sigma=1000$ for all of our experiments.
>
> Thank you for your minor comments. We will incorporate them in the final version!

---

> > ### Author Rebuttal · Reviewer_Br6q · 2026-04-03
> >
> > I thank the author(s) for a careful rebuttal which addresses all my concerns. I am happy to keep the score that I assigned to the paper.

---

### Official Review · Reviewer_3r8z · 2026-03-13

**Soundness:** 3
**Presentation:** 3
**Significance:** 3
**Originality:** 3
**Overall Recommendation:** 4
**Confidence:** 4

**Summary:**

The algorithm is based on work by Jourdan+, which gives the following coreset-based spectral clustering algorithm. They use an equivalence between kernel k-means and spectral clustering to show that a coreset designed for kernel k-means could be used to significantly accelerate spectral clustering. Basically, they extract a weighted kernel k-means problem from the graph and construct an reweighted induced subgraph (the coreset) on which they solve the normalised cut problem. Finally, they use this labeling to infer an accurate labeling for the rest of the original, massive graph. To build this coreset, they perfrom multiple rounds of $D^2$-sampling, and use a "sampling tree" data structure to efficiently pick representative nodes.

The authors of the current paper build directly upon static coreset algorithm, adapting it a fully dynamic setting where edges are added or deleted over time. The main idea is what they call "just-in-time sampling trees". The idea is simple: based on the previous algorithm, they figure out how these values $f(x,C)$ and $f(S,C)$ -- and the analogous versions with function $g$ -- change as edges are added and deleted, and their tree does not store these explciit values, but instead auxiliary info that allows them to reconstruct the values they need. Since the tree is balanced, they show that each update requires them to change only $O(\log n)$ values. Now the query time depends on the square of the maximum degree (times $k \log n$), plus a $\gamma(n, \epsilon, k, Y)$ term if you query a set $Y$. For the offline (static) setting, the query time was $O(nk \log n) + \gamma$, and so this makes sense for settings where $d_{\max}$ is much smaller than $\sqrt{n}$.

**Compliance With Llm Reviewing Policy:**

Affirmed.

**Final Justification:**

My concerns have been handled, I am happy with the paper.

**Key Questions For Authors:**

1. Can you relax the query time to depend on the average degree of the graph? Or even reduce the dependence on the maximum degree to just linear? If not, are there barriers or lower bounds?

2. Would your ideas be useful for other algorithmic problems?

3. The volume(Y) term in the gamma expression means this improvement is for sets with small volume. Is this a reasonable use case for the spectral clustering algorithms?

**Limitations:**

yes.

**Strengths And Weaknesses:**

The paper takes the offline result of Jourdan et al and extends it to the dynamic case; the ideas are not surprising, but are well-executed and give a reasonable query time when the graph has small maximum degree. It would be interesting to relax this query time, at the expense of slightly more sophisticated update operations.

The paper is reasonably well-written, though there are some mistakes. E.g., Theorem 1 talks about $d_{\max}$ but that seems to belong to Theorem 2, etc. The problem is interesting, and the work is original enough.

---

> ### Author Rebuttal · Authors · 2026-03-27
>
> Thank you for your comments and questions.
>
>
> **Weaknesses**
>
> Weakness 1: There are some mistakes. E.g., Theorem 1 talks about $d_{\max}$
>  but that seems to belong to Theorem 2
>
> We are not sure what you are referring to. Theorem 1 uses the average degree, $d_{\text{avg}}$, and Theorem 2 uses $d_{\max}$. We would be happy to correct any other errors you identify.
>
> **Questions**
>
> Q1: Can you relax the query time to depend on the average degree of the graph? Or even reduce the dependence on the maximum degree to just linear? If not, are there barriers or lower bounds?
>
> As mentioned in our response to reviewer Poyn, the second factor of $d_{\max}$ is very conservative. It assumes that all the neighbours of all the seeds that we sample belong to different seed sets. For many graphs, this is likely impossible. Existing dynamic sparsification techniques may also be employed to reduce the maximum degree too. See our response to Poyn for more context.
>
> Q2: Would your ideas be useful for other algorithmic problems?
>
> Yes! Indeed our main idea can be seen more as a dynamic graph clustering framework:
>
> - Any graph clustering algorithm can be used to cluster the coreset graphs, the theory simply assumes that the clustering algorithm gives an $\alpha$-approximation for the normalized cut problem given in (16).
>
> - Because our data-structures are updated in a lazy fashion, we can support richer edge types. One promising application is edges whose weights decay over a support interval (an edge is present in the graph in $[t_1,t_2)$ with weight decaying over time). This would allow practitioners to uncover relationships that snapshot clustering is incapable of. For example, transaction networks where edge weights correspond to exponentially decaying transaction sums.
>
> - The underlying just-in-time sampling idea works because we can represent the $D^2$-sampling distribution so that updates are sparse. It may be possible to apply the same technique to other problems where updates to the sampling distribution can be formulated in a sparse way.
>
> Q3: The volume(Y) term in the gamma expression means this improvement is for sets with small volume. Is this a reasonable use case for the spectral clustering algorithms?
>
> As mentioned in our response to Poyn, we foresee that the main application for this would be monitoring the relative cluster-membership of a fixed set of nodes over time. Maybe to identify when a previously formed community breaks down. Another example is feature extraction for downstream tasks that benefit from knowing whether nodes are from the same cluster or not. In this case, queries correspond to a small set of pairs of nodes.

---

> > ### Author Rebuttal · Reviewer_3r8z · 2026-04-02
> >
> > Sorry I was misreading, it seems, there is no d_{avg} and d_{max} issue. I am happy and will keep my score.

---

### Official Review · Reviewer_Poyn · 2026-03-13

**Soundness:** 3
**Presentation:** 2
**Significance:** 2
**Originality:** 2
**Overall Recommendation:** 4
**Confidence:** 3

**Summary:**

In the dynamic graph clustering problem, one is given a sequence of weighted graphs $G_1, G_2, \ldots$, where $G_i$ and $G_{i+1}$ differ by a single edge update. The goal is to design a data structure with minimal update time and query time, where a query asks for the clustering memberships of a specified set of vertices. The paper presents a dynamic data structure that maintains an $\alpha(1+\varepsilon)/(1-\varepsilon)$-approximate normalized cut. The structure supports edge insertions and deletions with an update time of $O(\log n)$, and answers set queries in time $O(k d_{\max}^2 \log n + \gamma(n,\varepsilon,k,Y))$, where $Y$ denotes the queried node set and $k$ is the number of clusters and $\gamma$ is some relatively complicated function.

The main contribution can be viewed as a dynamization of the static coreset spectral clustering algorithm of Jourdan et al. (2025).

**Compliance With Llm Reviewing Policy:**

Affirmed.

**Final Justification:**

The paper contains some interesting results, and I have raised my score to weak accept.

**Key Questions For Authors:**

-- Suppose we are only interested in the clustering memberships of a fixed set of vertices $Y$ and/or for a fixed value of $k$. Can your algorithm be modified to achieve a significantly smaller query time in this setting?

-- Why not simply write $\alpha(1+\varepsilon)$ instead of $\alpha(1+\varepsilon)/(1-\varepsilon)$? When $\varepsilon$ is small, we have $(1+\varepsilon)/(1-\varepsilon) = 1 + O(\varepsilon)$, so the expression could be simplified without changing the asymptotic guarantee.

-- Lines 203–204: please give the full name of **CSC** when it first appears.

-- Could you provide some intuitive explanation for the $O(\mathrm{vol}(Y) + d_{\max}^2 k)$ term in the running time? In particular, is it possible to reduce this dependence to $\tilde{O}(|Y|)$?

-- Currently, the fully dynamic algorithms appear to be interwoven with the analysis. I would suggest separating the algorithmic description from its analysis to make the presentation easier to follow.

**Strengths And Weaknesses:**

**Strengths:**

1. Studying dynamic coresets for spectral clustering is an interesting and relevant problem.

**Weaknesses:**

1. The theoretical guarantee appears somewhat complicated, and the query time has a rather involved form. It would help if the authors instantiated their bounds under several meaningful parameter regimes to better illustrate the implications of the result.

2. In addition, the query time depends linearly on $\mathrm{vol}(Y) + d_{\max}^2$, which can be quite large. This may limit the practical applicability of the algorithm. It would therefore be helpful to justify why such a query-time dependence is necessary.

3. The statement of the main theorem and parts of the algorithmic description are difficult to follow. For example, in the main results (Theorem 2 and Theorem 6), it should be made explicit that the normalized cut is a quantity that depends on $k$. Moreover, some key algorithmic details appear to be hidden in the analysis (for instance, around Line 244 (second column) ) which makes it harder to understand the dynamic algorithm itself.

---

> ### Author Rebuttal · Authors · 2026-03-27
>
> Thank you for your comments and questions. We think this work is an exciting contribution to the area of dynamic graph algorithms and hope our response below will convince you of its merits.
>
> **Weaknesses**
> 1. That's a great suggestion. There are multiple meaningful parameter regimes, which ones would you like highlighted? For example, when $d_{\max},|Y|$ and $k$ are all $O(\text{polylog}(n))$, our query time becomes $T_{NC}(\epsilon^{-4},\tilde{O}(1)) +\tilde{O}( \epsilon^{-8})$.
>
> 2. We provide an intuitive explanation for this term below. In theory, our bounds are very conservative but in practise, as demonstrated by our experiments, our algorithm is very fast.
>
>
> 3. Thank you for this. We will make it clear than the noramlized cut depends on $k$. For your second point, do we need more details on sampling trees, or are you referring to something else? We give full details of the proposed algorithm in Appendix A. We will highlight this in the main text.
>
> **Questions**
>
> 1. Suppose we are only interested in the clustering memberships of a fixed set of vertices
>  and/or for a fixed value of $k$. Can your algorithm be modified to achieve a significantly smaller query time in this setting?
>
> When you say fixed here, do you mean that the value is constant w.r.t $n$?  If $\text{vol}(Y)$, $k$ and $\epsilon$ are constant w.r.t $n$, our query running time collapses to  $\tilde{O}(d_{\max}^2)$. The critical observation is that the size of the coreset graph becomes constant, in which case we can solve the normalized cut problem exhaustively in constant time. This also means that the approximation ratio of the normalized cut algorithm becomes $\alpha=1$.
>
> The regime where $|Y|=O(1)$ is meaningful for monitoring the relative cluster-membership of a fixed set of nodes over time. Maybe to identify when a previously formed community breaks down.
>
>
>
> 2. Why not write $\alpha(1+O(\epsilon))$ instead of $\frac{\alpha(1+\epsilon)}{(1-\epsilon)}$ in the approximation guarantee?
>
> Yes, that will simplify our notation. We will change it in the final version.
>
> 3. Lines 203–204: please give the full name of CSC when it first appears
>
> Thanks for spotting that, we will add the full name there.
>
> 4. Could you provide some intuitive explanation for the
> $O(\text{vol}(Y) + d_{\max}^2k)$ term in the running time? In particular, is it possible to reduce this dependence to $\tilde{O}(|Y|)$?
>
> The $d_{\max}^2k$ part comes from running our efficient $D^2$ sampling routine. At each of the $O(k)$ steps in $D^2$ sampling, we choose the next seed vertex and it takes us $O(d_{\max}^2)$ time to update our data-structures to reflect this. If $k$ is constant and $d_{\max}^2 = O(\text{polylog}(n))$, then your assertion is correct.
>
>
>
> The factor of $d_{\max}^2$ is very conservative. It assumes that all the neighbours of all the seeds that we sample belong to different seed sets. For many graphs, this is likely impossible.
>
> 5. Currently, the fully dynamic algorithms appear to be interwoven with the analysis. I would suggest separating the algorithmic description from its analysis to make the presentation easier to follow.
>
> We will do this in the final version.

---

> > ### Author Rebuttal · Reviewer_Poyn · 2026-04-03
> >
> > Thanks for your response. There was one question which is not fully resolved.
> >
> > By “Suppose we are only interested in the clustering memberships of a fixed set of vertices and/or for a fixed value of $k$,” I mean the following. Suppose we are given a subset $S$ of vertices, where $|S|$ may be $O(1)$, $\mathrm{poly}(\log n)$, or $O(\sqrt{n})$, and suppose $k \le n^{0.01}$ throughout all the updates.
> >
> > In this setting, we would like to efficiently answer queries of the form: for any two vertices $u, v \in S$, do they belong to the same cluster at a given timestamp? What would be the query time of your algorithm under this restricted scenario?

---

> > > ### Author Response · Authors · 2026-04-03
> > >
> > > Thank you very much for the clarification. Upon further investigation, we found that we can tighten the dependence on $Y$ in the query time. The key observation is that we only care about edges between the queried set, $Y$, and the coreset vertices; $vol(Y)$ is a coarse upper bound for this. When $Y$ is large, a better bound is just the size of the coreset times the size of $Y$. So the component of our query time that depends on $Y$ reduces to
> > > $\min(|Y| \cdot \text{size of the coreset}, vol(Y)) = \min(|Y| \cdot k^2 /\epsilon ^4, vol (Y))$ - this is without any further assumptions on $Y$. We will add this in the final version of the paper.
> > >
> > > For your specific example this would become $O(\min (|Y|\cdot n^{0.02}, vol(Y)))$ (assuming $\epsilon$ is constant). If your queries are always of the form $(x,y)$ then the size of the queried set becomes a constant and the dependence on $Y$ in the query time collapses to $O(\min(n^{0.02}, d_{\max}))$. Unfortunately, the other components of the query time will remain unchanged because even if $Y$ is fixed and known in advance it is not obvious how to use this to speed up the coreset sampling part. This can an exciting direction for future work.

---

### Decision · Program_Chairs · 2026-04-30

**Decision:**

Accept (regular)

**Comment:**

The paper introduces a novel algorithm and efficient dynamic data structure for spectral clustering on dynamic graphs, achieving improved query times for maintaining approximate normalized cuts under edge and node updates.

Strengths
- Addresses the compelling problem of dynamic coresets for spectral clustering.
- Proposes a novel and interesting solution.

Weaknesses
-The experimental validation is somewhat limited in scope.

Decision
Accept (Poster)